# The Impact of Healthcare-Associated Infections in Patients Undergoing Oncological Microvascular Head and Neck Reconstruction: A Prospective Multicentre Study

**DOI:** 10.3390/cancers13092109

**Published:** 2021-04-27

**Authors:** Ana Ramos-Zayas, Francisco López-Medrano, Irene Urquiza-Fornovi, Ignacio Zubillaga, Ramón Gutiérrez, Gregorio Sánchez-Aniceto, Julio Acero, Fernando Almeida, Ana Galdona, María José Morán, Marta Pampin, José Luis Cebrián

**Affiliations:** 1Oral and Maxillofacial Surgery Department, “12 de Octubre” University Hospital, Institute for Biomedical Research (i+12), Universidad Complutense, 28041 Madrid, Spain; irene.urquiza@salud.madrid.org (I.U.-F.); ignacio.zubillaga@salud.madrid.org (I.Z.); ramon.gutierrez@salud.madrid.org (R.G.); gregorio.sanchez@salud.madrid.org (G.S.-A.); 2Unit of Infectious Diseases, “12 de Octubre” University Hospital, Institute for Biomedical Research (i+12), Department of Medicine, School of Medicine, Universidad Complutense, 28041 Madrid, Spain; flmedrano@salud.madrid.org; 3Oral and Maxillofacial Surgery Department, “Ramón y Cajal” University Hospital, Institute for Biomedical Research IRYCIS, Universidad de Alcalá, 28034 Madrid, Spain; juliojesus.acero@salud.madrid.org (J.A.); fernando.almeida@salud.madrid.org (F.A.); ana.galdona@salud.madrid.org (A.G.); 4Oral and Maxillofacial Surgery Department, “La Paz” University Hospital, Institute for Biomedical Research IdiPAZ, Universidad Autónoma, 28046 Madrid, Spain; mjose.moran@salud.madrid.org (M.J.M.); martamaria.pampin@salud.madrid.org (M.P.); josel.cebrian@salud.madrid.org (J.L.C.)

**Keywords:** head and neck surgery, reconstructive surgery, free flaps, healthcare-associated infections, surgical site infection, antibiotic prophylaxis, resistant microorganisms, osteoradionecrosis

## Abstract

**Simple Summary:**

Healthcare-associated infections (HAIs) result in an increased morbidity and a delay in adjuvant therapy—thus increasing the cancer recurrence rates—in patients undergoing oncological microvascular head and neck reconstruction. HAIs also result in a cost increase for the Health System. We prospectively analysed the incidence, clinical characteristics, risk factors and impacts of these infections in 65 patients undergoing head and neck free-flap reconstruction in three third-level university hospitals in Madrid (Spain). The three of them implemented the same antibiotic prophylactic regimen for surgical interventions. The rate of HAIs was 61.54%. The following complications were significantly more frequent in patients with HAIs: need to reoperate (*p* = 0.009), duration of hospital admission (*p* < 0.001) and delay in starting radiotherapy (*p* = 0.009). This manuscript aims to point out the importance of preventing HAIs in head and neck cancer patients, as they have shown a higher risk of postoperative complications.

**Abstract:**

(1) Background: Healthcare-associated infections (HAIs) after head and neck free-flap reconstruction are a common postoperative complication. Risk factors for HAIs in this context and their consequences have not been adequately described. (2) Methods: Ongoing prospective multicentre study between 02/2019 and 12/2020. Demographic characteristics and outcomes were analysed, focusing on infections. (3) Results: Forty out of 65 patients (61.54%) suffered HAIs (surgical site infection: 52.18%, nosocomial pneumonia: 23.20%, bloodstream infection: 13% and urinary tract infection: 5.80%). Methicillin-resistant *Staphylococcus aureus* (MRSA) and resistant *Pseudomonas aeruginosa, Klebsiella pneumoniae* and *Enterobacter cloacae* were the most frequently implicated. The significant risk factors for infection were: previous radiotherapy (Odds ratio (OR): 5.42; 95% confidence interval (CI), 1.39–21.10), anaemia (OR: 8.00; 95% CI, 0.96–66.95), salvage surgery (eight out of eight patients), tracheostomy (OR: 2.86; 95% CI, 1.01–8.14), surgery duration (OR: 1.01; 95% CI, 1.00–1.02), microvascular reoperation <72 h (eight/eight) and flap loss (eight/eight). The major surgical complications were: a need to reoperate (OR: 6.89; 95% CI, 1.42–33.51), prolonged hospital admission (OR: 1.16; 95% CI, 1.06–1.27) and delay in the initiation of postoperative radiotherapy (OR: 9.07; 95% CI, 1.72–47.67). The sixth month mortality rate in patients with HAIs was 7.69% vs. 0% in patients without HAIs (*p* = 0.50). (4) Conclusions: HAIs were common after this type of surgery, many of them caused by resistant microorganisms. Some modifiable risk factors were identified. Infections played a role in cancer prognosis by delaying adjuvant therapy.

## 1. Introduction

Nowadays, microvascular free-flap surgery is considered the ‘gold standard’ for the reconstruction of extensive three-dimensional defects on the head and neck after advanced cancer removal [1]. However, the high anatomical diversity and functional complexity of the head and neck determine the particularity of the reconstruction of this area, with a much higher variety of tissues to be reproduced than in any other part of the body. Therefore, microvascular reconstruction, as a complex surgical procedure performed in patients often with multiple comorbidities—as in oncological patients—can induce serious postoperative complications. The most frequent ones are healthcare-associated infections (HAIs) [1,2].

In this study, HAIs are defined according to the United States Centres for Disease Control and Prevention (CDC) guidelines [3]. In the clinical practice, based on the frequency of appearance, the four commonly considered types of HAIs are: surgical site infection (SSI), bloodstream infection, respiratory infection and urinary tract infection (UTI).

The incidence rate of HAIs in hospital patients is a quality and safety indicator of patient care, being the most common complication of unsafe patient care worldwide [3,4,5,6]. Specifically, in patients undergoing microvascular head and neck reconstructions, the incidence of HAIs—headed by SSI [5]—is estimated as up to 44% [7,8,9,10,11]. Besides, HAIs differ from community-acquired infections in the microbiological pattern, and usually, the microorganisms involved tend to have higher rates of antibiotic resistance.

Infections may lead to free-flap complications (necrosis or delay wound healing) potentially worsening the patient’s condition when swallowing, speaking or breathing; aesthetic result and, finally, poor quality of life. As a consequence, hospital stays are prolonged on many occasions with high morbidity rates [6]. For this reason, apart from being an added complication for the patient—sometimes serious or even fatal—they involve an economic over-cost [6,12,13]. Furthermore, HAIs also play a role in cancer prognosis. They can increase the risk of tumour recurrence and mortality rates due to the inflammatory response produced [2], also causing a delay to adjuvant therapy [2,14,15].

Another fact to consider in head and neck surgery is the difficulty and controversy when establishing an antibiotic prophylaxis regimen [12,16]. Oral cavity oncological surgery and, in general, huge head and neck defects are classified as clean-contaminated according to the CDC classification [3,4]. However, the incidence of HAIs among patients with free flaps is even higher compared to other types of surgeries of this group. This could be explained by the complexity and length of microvascular reconstruction, as well as the large oncological resections that usually communicate the neck and the oral cavity, thus exposing cervical tissue to contaminated saliva and secretions from the respiratory and digestive tracts [17]. Nowadays, there is no consensus on the selection of antibiotics and whether to recommend them for or against the use of long-term antibiotic prophylaxis duration [5,7,9,18,19], due to the high incidence of patients suffering HAIs and the emergence of multidrug-resistant (MDR) bacteria.

In this manuscript, we aim to measure the impact of HAIs in patients undergoing free-flap reconstruction on the head and neck and identify the potentially modifiable risk factors.

## 2. Materials and Methods

### 2.1. Study Design

We performed an observational prospective multicentre study developed in the Oral and Maxillofacial Surgery Department of 3 Spanish institutions: ‘12 de Octubre’, ‘Ramón y Cajal’ and ‘La Paz’ University Hospitals. These are tertiary-level hospitals with 1000–1400 beds (three of the seven largest hospitals of Madrid and three of the ten largest ones in Spain).

Inclusion criteria were: (1) patients older than 16 years old and (2) microvascular free-flap reconstruction of head and neck surgery performed between 1 February 2019 and 31 December 2020. The surgeries were part of the treatment of benign tumours, malignant tumours or osteoradionecrosis (ORN) deformities. Patients underwent surgery following the decision of the Multidisciplinary Head and Neck Tumour Board. Locations included: oral cavity, oropharynx, skin, skull base and paranasal sinuses.

We excluded patients when surgery was inadvisable because of systemic pathologies or with unresectable tumours without a palliative option.

### 2.2. HAIs Strategy and Diagnostic Method

The principal preoperative strategies for preventing HAIs in head and neck reconstructive surgery at the three hospitals were:(1)Hair removal immediately before surgery by clipping [20].(2)Preoperative antiseptic showering with chlorhexidine soap [20].(3)Same prophylactic antimicrobial regimen for all patients with Amoxicillin/clavulanic acid 2 g or Gentamicin 120 mg with Clindamycin 600 mg if allergic. Antibiotics were started 30–60 min before incision and continued every eight hours for three days after surgery for benign tumours or ORN deformities, five days for primary or secondary malignant tumours or seven days for salvage surgeries.(4)Disinfection of the surgical field with 2% chlorhexidine for the skin and 0.5% chlorhexidine for the oral cavity.

As previously introduced, the criteria used to define HAIs followed CDC guidelines [3]. When suspected, a specimen was obtained from the potentially infected field and was cultured for microorganisms. Antimicrobial susceptibility testing was performed at each participating centre according to the methodology established by the European Committee on Antimicrobial Susceptibility Testing (EUCAST). MDR was defined by testing for acquired resistance to at least one agent in more than three different categories of antibiotics [21]. Empirical antibiotic treatment was stared at the same time, once antimicrobial susceptibility testing was obtained, with a change of antibiotic if necessary. Treatment was supervised by the Infectious Diseases Treatment Unit.

### 2.3. Study Definitions/Variables

The following preoperative variables were collected to describe the sample:(1)Demographic characteristics: age and sex.(2)Comorbidities: diabetes mellitus, immunosuppression, active tobacco use, previous head and neck surgeries or radiotherapy (RT), Charlson comorbidity score [22], hospital admission in the last two years and previous colonisation by resistant microorganisms.(3)Laboratory disorders: anaemia—defined as haemoglobin <11.4 g/dL for females or <13 g/dL for males—, hypalbuminaemia—considered when albumin <3.5 g/dL—, hypo- or hyperleukocytosis—defined as WBC <4 or >11.3 x109/L— and C-reactive protein (PCR) >0.5 mg/L.(4)Nutritional status: height, weight and BMI.

Perioperative factors included:(1)American Society of Anaesthesiology (ASA) score.(2)Preoperative diagnosis: benign tumour, primary malignant tumour, secondary malignant tumour, salvage surgery or ORN deformities.(3)Duration of surgery.(4)Tracheostomy.(5)Neck dissection: side and type.(6)Location of the oncological resection: oral cavity, oropharynx, skin, skull base or paranasal sinuses.(7)Wound classification: clean, clean-contaminated, contaminated or dirty [16].(8)Type of free flap: ALT, forearm, fibula or others.(9)Need to reoperate in the first 72 h, including flap loss.

In the three institutions, a close monitoring of the flaps during the first 72 h was carried out every 2 h, consisting of: direct evaluation of the flap characteristics: skin colour (normal/cyanotic/ischemic), temperature (hot/cold) and capillary refill (1–3 s/3–5 s/more than 5 s); handheld Doppler signal; pinprick test when necessary (arterial bleeding/venous bleeding/no bleeding) and neck evaluation (normal/oedema/tension hematoma) [23,24].

Infection characteristics involved were:(1)Type of HAI: donor or recipient SSI, respiratory infection (nosocomial pneumonia or tracheobronchitis), bloodstream infection (with or without sepsis) or UTI.(2)Microorganism identified.(3)Treatment received (antibiotic, posology and duration).

Regarding postoperative complications, we included: a need to reoperate after 72 h, increased duration of hospital admission, 30-day readmission rate, delay of adjuvant RT, mortality rate and disease-free survival (DFS).

### 2.4. Data Collection Methods

Study data was collected and managed anonymously using Research Electronic Data Capture (REDCap), where the principal researcher of each hospital was in charge of introducing their patient’s data, with restricted access for the rest of the collaborators. All data were gathered from the patients’ electronic medical records (EMR), based on available notes from the teams involved.

### 2.5. Statistical Analysis

Quantitative variables were defined through the median (p50) and the interquartile range (p25–p75). The Shapiro–Wilk test was used to check the normal distribution. Qualitative variables were expressed in absolute numbers (number of cases) and in relative frequencies (percentage). The relationship between the variables and HAIs was assessed using chi-square test or Fisher’s exact test for qualitative variables. Quantitative variables were compared using the Mann–Whitney *U* test. Additionally, the statistical study was completed with a multivariate analysis using a binary logistic regression to analyse the factors associated with a higher risk of HAIs and complications. Odds ratios (OR) with 95% confidence intervals (95% CI) were estimated. The analysis was performed using the SAS for Windows version 9.4 statistical software (Copyright © 2021–2012 by SAS Institute Inc., Cary, NC, USA). In all cases, the level of confidence was 95%, considered a statistical significance when *p* ≤ 0.05.

### 2.6. Ethics Approval

The study protocol was endorsed by the Ethics Committee for Clinical Research of the University Hospital “12 de Octubre” and by the local Ethics Committees of the other centres, the Pharmaceutical Control and Health Products Area of the Community of Madrid and the Spanish Agency for Medications and Healthcare Products. We conducted it in accordance with the ethical standards of the 1964 Declaration of Helsinki and its later amendments [25].

## 3. Results

### 3.1. Cohort Study

In this study, we included 65 patients from the three institutions (Table 1), 41 males (63.08%) and 24 females (36.92%). The median age was 64 years (range: 18–85), and 60.00% of them were between 50–70 years old. The most common Charlson Comorbidity Index (CCI) was four, predicting a ten-year survival of 53.00%. More than half of the patients were tobacco users (53.85%), 23.08% underwent head and neck surgery before and 30.77% previously received RT. The most common ASA score was III (47.69%), followed by II (41.54%), I (7.69%) and several IV (3.08%).

Forty-one patients (63.08%) underwent microvascular reconstruction as part of the treatment for primary malignancies, 12.31% after a secondary tumour on the head and neck, 10.77% after salvage surgery procedures, 7.69% due to ORN mandibular reconstruction and 6.15% after the resection of benign tumours. Some (63.08%) of the patients needed a tracheostomy. Neck dissection was performed in 64.62% of the cases, usually unilateral (69.05%) and for selective levels I-III (71.43%). Oncological resection was performed mostly in the oral cavity (76.92%), also including the oropharynx, skull base, skin and paranasal sinuses (7.69%, 6.15%, 6.15% and 3.08%, respectively). Microvascular head and neck reconstruction was performed using an anterolateral thigh flap (ALT) in 33.85% of the cases, fibula (29.23%) and forearm flap (29.23%). The median duration of the surgery procedure was 600 min (IQR 510-670).

During admission, 26 patients (40.00%) had to be reoperated on. Treatment outcomes for each surgery are detailed in Table 1. Four patients (6.15%) needed more than one intervention. A total of 36 surgeries were needed.

Surgeries performed in the first 72 h were directly related to the microvascular surgery procedure and involved anastomoses review (due to arterial or venous thrombosis) and hematoma drainage. In our series, 13 patients returned to the operating room in the first 72 h. Seven out of the eight patients in whom vascular anastomoses were performed again suffered flap loss (87.50%). Four of them underwent another reconstruction during hospital admission, using free flaps in two patients (forearm) and regional flaps (pectoralis and temporal) for the other two. In the other three patients, reconstruction of the defect was delayed and performed using free flaps (one fibula, one forearm and one ALT, respectively). By contrast, for the other four patients who underwent hematoma drainage without redoing anastomosis, the flap survival rate was 80%. Other surgeries performed after the first 72 h were needed in 17 cases (26.15%). Dehiscence closure with or without fistula closure (43.48%) was the most surgery type, followed by abscess drainage (30.43%) and donor surgical site complications (8.70%). These surgeries were directly related to HAIs.

The median duration of hospital admission was 19 days (IQR 12–26, range: 6–101). For 81.54% of the patients, the stay was ≤one month. The 30-day readmission rate was 14.29%. For 77.77% of the cases, the cause was infectious (cervical abscess or pneumonia).

Thirty-three patients (50.77%) received adjuvant RT (72% of the primary malignant tumours). The median treatment start time after surgery was eight weeks (IQR six–nine). In 13 patients (39.39%) RT started during the seven weeks after surgery (the median was six weeks, IQR six to seven). However, in 60.61% of the patients, RT was delayed more than seven weeks (the median was eight weeks, IQR 8–11) in 80.00% of the cases due to HAIs (active infection, prolonged hospital admission, delay of wound healing, etc.).

The six-month overall survival (OS) was 92.86%. Only two patients died, one within the first month after surgery and the other one after two months, both due to respiratory failure after pneumonia. The one-year OS was 72.73%—76.92% for oncological patients—and DFS in the first year resulted in 68.18%.

### 3.2. Characteristics of Healthcare-Associated Infections

The incidence of HAIs among patients undergoing microvascular free-flap head and neck reconstruction was 61.54% (Table 2). Eleven out of these 40 patients (27.50%) suffered from one infection, whereas 29 patients (72.50%) had more than one infection, raising the total number of HAIs to 69.

SSIs were the most frequent HAI with 52.18%—46.38% corresponding to recipient SSI and 5.80% to donor SSI—, followed by nosocomial pneumonia (23.19%), bloodstream infection (13.04%) and UTI (5.80%). Regarding infections in the recipient surgical site, the neck was the most common location (78.12%). The rest of the remaining four correspond to the face (15.63%) and oral cavity (6.25%). Donor SSIs (four cases) were treated conservatively with antibiotics, but one case (25%) required another surgery for placing a new graft at the donor area. Respiratory infections were present in 16 patients (24.61%), nine developing pneumonia with radiological signs—two requiring Intensive Care Unit (ICU) management—and the other seven with a positive tracheostomy exudate culture without radiological signs of pneumonia (tracheobronchitis). One out of the nine patients had bacteraemia developed sepsis. The four UTIs were treated conservatively with a pharmacological treatment. Different additional HAIs that were found included gastrointestinal infections (three) and one abdominal abscess after performing a gastrostomy.

With regard to the microbiological data, the majority of isolated microorganisms were bacteria (90.00%). We only found viruses in three cases (4.28%), fungi in four cases (5.71%) and no parasites. *Staphylococcus, Pseudomonas, Klebsiellas* and *Enterobacter* species were frequently cultured. All cultured bacteria are detailed in Table 3. The rate of MDR bacteria was 36.62%.

The most commonly used antibiotics for the treatment of HAIs were meropenem 1000 mg every 8 h (h) intravenously (IV); piperacillin–tazobactam 4000 mg/500 mg every 6 h IV; amoxicillin–clavulanic acid 2000 mg/200 mg every 8 h IV and ciprofloxacin 500 or 750 mg every 12 h IV or oral administration (used in 52.50%, 32.50%, 30% and 30% of the patients, respectively). Other commonly used antibiotics were tigecycline, with a loading dose of 100 mg followed by 50 mg every 12 h IV, linezolid 600 mg every 12 h IV or oral therapy and ertapenem 1000 mg every 24 h IV (used in 22.50%, 20% and 20% of the patients, respectively). Some (67.50%) of the patients needed more than one antibiotic for HAI or HAIs treatment or due to MDR bacteria. The median duration of the antibiotic therapy was seven days (IQR 5–10).

### 3.3. Risk Factors for HAIs

In our series, the age and gender distributions between patients with and without infection were similar (Table 1). All comorbidities analysed, altered presurgical blood parameters and pathological body mass index (BMI) were more frequent among patients with HAIs. By a univariate analysis, we found statistically significant differences in patients with a history of previous RT on the head and/or neck, with a 5.42 higher risk of suffering from infection (Table 4) (OR 5.42; 95% CI, 1.39–21.10; *p* = 0.013). Concerning the blood parameters, anaemia was present in 25% of the patients with HAIs, resulting in eight times more likely to suffer HAIs in patients with anaemia (OR 8.00; 95% CI, 0.96–66.95; *p* = 0.040). We did not find any statistical differences among the nutritional parameters (hypalbuminaemia and BMI < 18.5 or >25).

Regarding the surgical factors, we observed a significant difference in the distribution of the diagnosis prior to microvascular reconstruction (*p* = 0.028). As shown in Table 1, 100% of patients undergoing salvage surgery presented at least one HAI (*p* = 0.038). Tracheotomised patients also had higher rates of infection (72.50% vs. 48.00%; OR 2.86; 95% CI, 1.01–8.14; *p* = 0.065). The median surgery duration was one hour and 30 min longer in the group that suffered HAIs (OR 1.01; 95% CI, 1.00–1.02; *p* = 0.034). All patients who needed microsurgical revision of the anastomoses in the first 72 h—and closely linked to this, all patients with flap loss—suffered infection (*p* < 0.001 and *p* = 0.038, respectively). In contrast, the HAI incidence in patients who underwent hematoma drainage in the first 72 h and/or those with flap survival was similar to the total cohort incidence (60.00%).

A multivariate logistic regression analysis revealed that previous RT on the head and neck (OR 5.57; 95% CI, 1.41–22.69; *p* = 0.014) and a longer duration of surgery (OR 1.01; 95% CI, 1.001–1.01; *p* = 0.048) were independently associated with higher rates of infection (Table 4).

### 3.4. Analysis of Complications in Patients with Head and Neck Free-Flap Reconstruction

According to Table 1 and Table 5, the incidence of complications discussed in our study in the group of patients with HAIs was significantly higher (97.50% vs. 32.00%; OR 17.34; 95% CI, 2.09–143.28; *p* < 0.001). Only one patient with a HAI did not require additional surgeries; his hospital admission was <two weeks, he was not readmitted in the first month, he received RT within seven weeks after surgery and he is alive. On the contrary, 68% of the patients without infection did not suffer from any of the analysed complications.

Fifteen patients with HAIs (37.50%) needed to be reoperated on after the first 72 h of microvascular surgery, whereas only two patients without HAI (8.00%) had to (OR 6.89; 95% CI, 1.42–33.51; *p* = 0.009). The duration of hospital admission was much longer for patients with HAI (23 days vs. 11 days; OR 1.16; 95% CI, 1.06–1.27; *p* < 0.001). The 30-day readmission rate in the cohort was 14.29%: 17.95% in the group of patients with HAI vs. 8.33% in patients without HAI (*p* = 0.46).

Adjuvant RT was indicated in 55% of the patients with HAI and 44% of the patients without HAI. A delay in the beginning of RT after surgery (>seven weeks) among patients with HAI was nine times more frequent than in patients without infection (OR 9.07; 95% CI, 1.72–47.67; *p* = 0.009). The number of weeks from surgery to the initiation of adjuvant RT between the two groups was also statistically different (eight weeks (IQR 7.50–10.50) vs. six weeks (IQR 5–7); OR 2.94, 95% CI, 1.25–6.93; *p* = 0.002). Both deaths in the first six months after surgery occurred in patients who suffered an infection, the cause of death being nosocomial pneumonia.

The Cox regression model pointed out that a prolonged duration of hospital admission (OR 1.16; 95% CI, 1.05–1.28, *p* = 0.002) and a delay in the start of adjuvant RT (OR 12.00; 95% CI, 1.89–76.38; *p* = 0.008) had significant negative impacts on patients undergoing free-flap head and neck reconstruction, regardless of infection (Table 5).

## 4. Discussion

Patients undergoing major head and neck surgical procedures are highly predisposed to develop postoperative complications. HAIs seem to be the most frequent one in the literature data [7,8,9,10,11]. To the best of our knowledge, the present study represents the largest prospective cohort assessing microbiological data, clinical characteristics and the outcomes of all types of HAIs among patients undergoing microvascular free flap in the head and neck region and the first study that implements a standardised antibiotic prophylaxis regimen. Several relevant conclusions about the impact of HAIs in these patients can be derived from our series.

### 4.1. Antibiotic Prophylaxis Regimes

There has been substantial controversy in the literature with respect to the recommended antibiotic regimen that should be used in head and neck surgery [12,16], even more controversial when it comes to microvascular reconstruction. In most articles, the prophylaxis guideline recommendations usually fail to address the complexity of microvascular reconstruction compared to other head and neck surgeries (duration of the procedure, oro-cervical communication, etc.) [26].

The extended use of perioperative antibiotics during more than 24 h is the most common guideline, as it seems to reduce the incidence of HAIs [1,5,9,18,27,28,29,30,31]. Nevertheless, there is no clear standard for the number of days antibiotics should be held (Karakida et al. suggested three days [1], Reiffel et al. five days [18] and Saunders et al. seven days [9]).

Karakida et al. prolonged antibiotic therapy until the third postoperative day [1], Reiffel et al. up to the fifth day [18] and Saunders et al. until the seventh day [9]. When comparing the short-term (24-h regimen) vs. long-term (>24 h) use of antibiotics, the studies by Haidar et al. and by Bartella et al. both asserted that the risk of SSI was significantly higher in patients receiving prophylactic antibiotics ≤24 h compared to >24 h (OR 1.56 [27]), although the incidence of other nosocomial infections (pneumonia, UTI and sepsis) did not decrease under prolonged antibiotic prophylaxis [27,28]. On the other hand, certain authors encouraged short durations of broad-spectrum antibiotics, since long-term antibiotics did not appear to reduce the incidence of SSIs, with an increased risk of developing antibiotic resistance [7,19,26,31]. In summary, in the recent literature, no specific pattern of antibiotic prophylaxis has substantially proven to be effective.

It is worth mentioning that, in the three institutions of this study, all patients undergoing microvascular surgery received the same antibiotic prophylaxis protocol. In our proposed regimen, the duration of the antibiotics was based on the diagnosis prior to surgery, since the surgical complexity is generally linked to the diagnosis, and, in turn, to the risk of acquiring the infection intraoperatively. However, it should be clarified that, in order to establish the optimal antibiotic prophylaxis regimen and whether it should be different according to certain criteria, a comparative study using different regimens would be necessary.

### 4.2. Infection Characteristics

In the present cohort, the incidence of HAIs in microvascular free-flap head and neck reconstruction was 61.54%, and the incidence of SSI, with or without another infection, was 43.10% (52.18% of all HAIs). These figures, although relatively high, were consistent with the published literature [7,8,9,10,11].

The idiosyncrasy of head and neck free-flap reconstruction, which makes it more vulnerable to infection, is due to multiple factors. First, surgery is performed at different sites: tumour resection, neck dissection, tracheostomy and the donor site region, thus creating multiple surgical sites with variable local flora, which the patient has to deal with and the healthcare personnel must take care of. Moreover, tracheotomy and resections in the upper aerodigestive regions will disable or limit the patient’s ability to swallow correctly, which can be considered as the major cause for respiratory tract infections due to aspiration. Finally, the complex anatomy of the oro-cervical region and the use of distant microvascular free flaps for the reconstruction often do not allow water-tight closure of the surgical wound, as this potentially leads to prolonged contamination and fistula formation, especially in defects associating intra-oral and extra-oral communication.

Where most articles focus only on SSI, Bartella et al. collected all types of HAIs [28]. Comparing both results, the percentages of SSI and bloodstream infections in all HAIs were similar (SSI: 52.12% in our series vs. 63.33% in Bartella el al.’s; bloodstream infection: 13.04% vs. 16.66%, respectively); the incidence of pneumonia was noticeable in our series (10.14% vs. 0%) and lower for UTI (5.80% vs. 20%) [28]. In the series by Carniol et al., the SSI incidence was comparable (46.30%), also with more similar rates to ours in terms of pneumonia and UTI (6.60% and 1.20%) [8]. Several factors could explain the variation in incidence of the different types of HAIs. First, the heterogeneity of the sample with regards to the tumour location—oral cavity, oropharynx, skin, skull base and paranasal sinuses—each with distinctive flora and therefore, perhaps, different behaviours. Second, the tumour stage, which dictates the extent of surgical resection and influences the complexity of the microsurgical reconstructive technique. Finally, another potentially confounding factor may be the definitions of the HAIs and HAI types, which are sometimes clinically inconsistent [3,4].

As to the microbiological data, the species cultured in our series were similar to other studies [5,26,28,29]. From the entire list of microorganisms, Methicillin-resistant *Staphylococcus aureus* (MRSA) and MDR *Pseudomonas aeruginosa* stood out as the most common bacteria, both of which were cultured in 10.14% of all HAIs. A high incidence of HAIs by resistant microorganisms was also observed in the present study, with a rate of MDR bacteria of 36.36%. As the CDC noted, the overall antibiotic resistance is increasing, making the choice of prophylaxis and treatment very difficult [3,6]. Our percentage of MDR bacteria might be concerning, as it has already been demonstrated that, compared to susceptible infections, MDR HAIs result in an increased cost to the health system, a prolonged duration of hospital stay and excess in-hospital mortality [32].

### 4.3. Risk Factors for HAIs

In an attempt to curb the incidence of HAIs, we investigated the potential risk factors for HAIs in five categories: patient demographics, comorbidities, blood parameters, nutritional status and surgery characteristics. According to the CDC guidelines, the patient characteristics proven to be factors to increase the risk of HAIs are: age, nutritional status and diabetes [3]. However, in this study, none of them showed a statistically significant difference.

Several articles confirmed the association between malnourished patients—considered as those with a low Nutrition-Related Index (NRI) caused by hypoalbuminemia and altered BMI—and infection [29,33]. Serum albumin is a known biochemical marker for malnutrition in predicting perioperative complications in head and neck surgery [29,33], although the direct relationship between hypoalbuminemia and infection in free-flap reconstruction is unclear [29,30]. The BMI is the most common clinical indicator of the nutritional condition, and it is also related to inflammation, prevalent in the oncological population but independent of infection [33]. In-line with most authors, at the three institutions participating in this study, a nutritional assessment was also carried out prior to surgery to improve the peri-surgical nutritional status and, thus, wound healing.

The correlation between smoking and postoperative complications in free-flap reconstruction of the head and neck is of particular interest, since smoking represents a major risk factor for the development of head and neck cancer [34] and has demonstrated that it can disrupt the normal vascular physiology, which may compromise the integrity of the microvascular anastomosis. In the present analysis, we found that more than half of the patients who underwent head and neck free-flap surgery were active smokers. However, there was no difference in the infection rates among smoking patients.

A high percentage of patients with immunosuppression also stood out in our series, without being related to the infection rates. The role of the immune status in head and neck infections after major reconstructive procedures has not been extensively studied. Grandis et al. suggested that individuals with a greater degree of immunosuppression may be less able to respond adequately to bacterial contamination of the operative area, but they report needing more data to elucidate the elements of the immune system responsible for controlling the infection or tumour progression, so as to possibly develop therapies that specifically address host immunocompetence [15].

As expected, a high comorbidity rate was present in our cohort (CCI score ≥ 4), although it was not associated with higher infection rates. In an oncological surgery, the use of CCI to predict the general complications and mortality is very common [22,33,35], unlike for local complications, such as SSI [33,35].

Regarding the blood parameters, presurgical anaemia was confirmed to be a risk factor for HAIs. Several articles have linked blood loss during surgery and perioperative transfusion with a higher rate of wound infections [1,36], but none of them mentioned preoperative anaemia, which could be complementary. This may be of interest, since the haemoglobin correction prior to surgery could be an easily implementable measure.

Our experience confirmed that previous head and neck RT is a risk factor for infection (OR 5.42). Most authors agree that irradiated wounds elicit a poor inflammatory response in local complications, thus delaying the identification and treatment of abscesses, which ultimately leads to major infectious complications requiring additional surgeries [10,11,14,29,36]. This can also be derived from our results, as HAIs appeared in all salvage surgeries and 87.50% of the second malignancies, all with previous RT history. This might explain the high frequency of HAIs in our cohort, where secondary malignant tumours, salvage surgeries and osteoradionecrosis accounted for a total of 20 patients, 30.77% of the entire cohort. Moreover, these tumours often required larger resection areas, which increased the surgical stress, extended the surgical time and sometimes led to dead spaces in the head and neck that subsequently filled with blood clots or seromas and became superinfected. In light of the above discussion, and since previous RT in our cohort was proven to be a strong risk factor for infection, we may reconsider in the future the duration of our antibiotic prophylaxis regimen for ORN deformities.

Although most authors have claimed that free flaps themselves, as a reconstructive surgery technique, are a risk factor for HAIs in contrast to regional flaps, no differences between the flap types and infection rates were identified [12,36]. Concerning flap survival, our free flap failure rate (12.30%) was slightly higher than in the other series (5–11.00%) [23,24,35,37], while the flap survival rate after a cervical haematoma drainage (80%) and after salvage surgery for vascular thrombosis in the first 72 h was similar (12.50%) [37,38]. Regardless, all patients with flap failure suffered HAIs, but infection after salvage surgery was similar for the whole cohort (60%). In addition, the surgery duration in our series was also significantly related to the risk of infection, with a median duration of 1 h and a half longer for the group suffering HAIs. Other authors have also demonstrated this relationship in patients with free-flap head and neck reconstruction [37,39]. These two findings—a high rate of free-flap failure and prolonged duration of surgery—should be areas for improvement.

As previously mentioned, and in-line with the published literature [19], tracheotomised patients had higher rates of infection (72.50% vs. 48%), as their capacity to swallow is compromised, with the consequent risk of aspiration and, potentially, pneumonia.

### 4.4. Impact of HAIs on Surgical Complications in Patients Undergoing Oncological Microvascular Head and Neck Reconstruction

Postoperative complications after head and neck microvascular reconstruction—apart from HAIs and free-flap failure— arose in 47 patients (72.31%), as presented in Table 1. When comparing the adverse events, we identified a clear trend toward higher postoperative complication rates—especially for reoperation, prolonged hospital admission and delay of adjuvant RT—in patients that suffered HAIs (OR 17.34, 95% CI 2.09–143.28; *p* < 0.001).

The need for reoperation 72 h after microvascular surgery was significantly higher in patients with HAIs (OR 6.89, 95% CI 1.42–33.51; *p* = 0.009), the reason for surgery being closely linked to the infection (Table 1). In any case, reoperation has a significant added cost and considerably prolongs the length of the hospital stay, as shown in the present study. Al Qurayshi et al. studied the economic impact of SSI due to a prolonged hospital stay (8.1 +/− 0.8 days per case), resulting in an increase in the health service costs of $ 20,953 [13]. From Broome et al.’s work, it can also be concluded that postoperative complications (including HAIs) are a predictor of the length of hospitalisation [40]. Our median hospital stay (19 days) was similar or even slightly lower than in other centres [38]. While 57.50% of patients with HAIs had a median hospital stay of more than three weeks, only 12% of patients without HAIs remained in hospital that long (*p* < 0.001). This fact was often linked to the delay in wound healing or the duration of antibiotic treatment.

Nowadays, the 30-day readmission rate is a highly scrutinised metric of surgical care quality, since readmission results in further costs, and it is believed to be avoidable with planning and patient education [41]. Head and neck surgery patients generally share multiple risk factors for readmission, such as older age, added comorbidities, lower socioeconomic status and a history of multiple visits to the emergency department (as they often seek primary care there). Readmissions after oncological surgery can be detrimental to the patient’s prognosis, as they may delay the beginning of radiotherapy or chemotherapy, which has proven to increase mortality. As in other reports, infections (specifically, SSI and pneumonia) were the usual cause of readmission, and treatment for the readmission included surgery (drainage, debridement, reconstruction of the surgical site, etc.); wound care and intravenous antibiotics. Therefore, the evaluation of strategies to promote wound healing and prevent infections might reduce readmissions and increase the costs and morbidity associated with head and neck free-flap surgery.

The delay in adjuvant therapy is of key interest in assessing the impact of HAIs on cancer prognosis. The new European Clinical Practice Guidelines for the diagnosis, treatment and follow-up of squamous cell carcinoma of the head and neck declared that postoperative RT should be started within six or seven weeks after surgery and/or that the treatment regimen of surgery and postoperative RT should be delivered within 11 weeks [42]. According to this, we found that patients with postoperative infections were 9.07 times more likely to have a delay in starting RT treatment (>seven weeks), as indicated in Table 5. Although the cause of delay in our series was, in 80% of the cases, due to HAIs (active infection, prolonged hospital admission, delay of wound healing, etc.), there were also additional causes that also need to be taken into account to reduce any delay in starting adjuvant therapy—such as logistical issues in the Pathology/Radiation Oncology/Medical Oncology Departments, serious medical diseases of patients, etc. Grandis et al. detected that patients with SSI were 3.20 times more likely to develop a recurrent disease than patients with tumours of similar stages who did not experience an infection [15]. In addition, patients with wound infections were 2.40 times more likely to die from the disease than their infection-free counterparts [15]. Furthermore, the infection itself has also been studied as a cause of an increased recurrence rate due to the released inflammatory response [2]. Perioperative mortality in our series found no statistically significant differences, probably due to the small sample size, although the two deaths in the first six months after surgery occurred in patients who suffered from infection, the cause of death being infectious.

In brief, the relationships between all postoperative complications—including HAIs—are presented in Figure 1. As illustrated, most of the complications fell into the large circle of infections in two main groups: on the left-hand side, patients reoperated on in the first 72 h related to microvascular surgery and, on the right-hand side, patients reoperated on after 72 h principally due to infection development. Accordingly, patients with a hospital admission of more than three weeks, patients who were readmitted 30 days after discharge, those with a delay in the start of adjuvant RT after surgery and/or those who died are grouped together and within the main circle of patients with HAIs.

### 4.5. Study Limitations

The present study had one main limitation: the small sample size. For this reason, when patients were classified into subgroups, the sample size was occasionally insufficient to generate statistically significant conclusions. These results should then be interpreted with particular precaution due to the reduced sample of the cohort.

## 5. Conclusions

In summary, the clinicians in charge of patients undergoing free-flap reconstruction of the head and neck should beware the high frequency with which these patients suffer HAIs, as they face a significantly higher risk of postoperative complications. We consider it essential to recognise the factors that predict the HAIs and to take appropriate perioperative measures to minimise wound complications and to prevent any delay in adjuvant treatment, which is associated with increased costs to the hospital. This ongoing study should, in the future, better clarify the impact of infections on cancer recurrence and long-term survival by delaying adjuvant treatment.

## Figures and Tables

**Figure 1 cancers-13-02109-f001:**
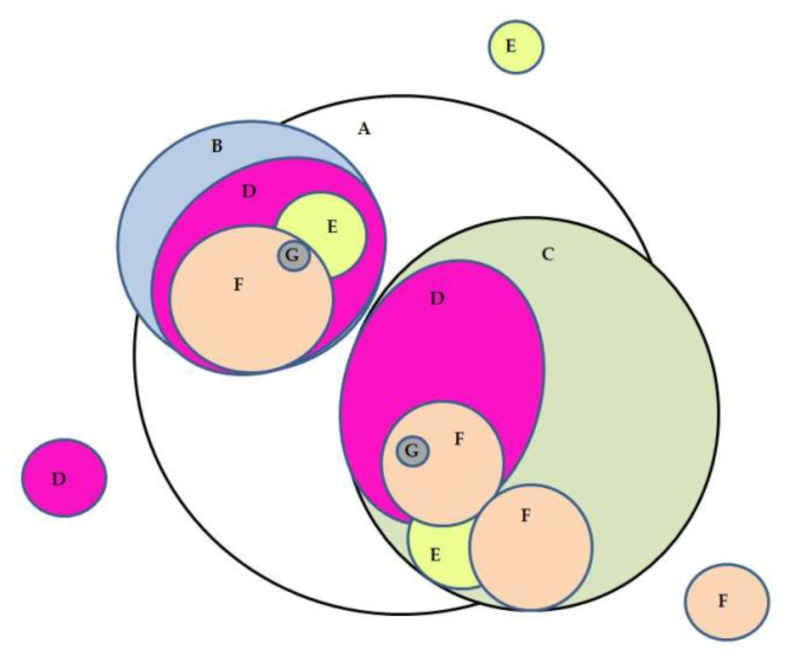
Relationship between postoperative complications. The size of the circles is proportional to the number of patients suffering from the described complications, the complications in order of appearance from the inside out. (A) HAIs, healthcare-associated infections. (B) Reoperation <72 h. (C) Reoperation >72 h. (D) Hospital admission >3 weeks. (E) Readmissions. (F) Delay in adjuvant radiotherapy (RT). (G) Deaths.

**Table 1 cancers-13-02109-t001:** Descriptive analysis for HAIs after microvascular reconstruction on the head and neck for oncological patients (February 2019–December 2020).

	Total	Patients with HAIs	Patients without HAIs	*p*-Value
Variable	65	40 (61.54%)	25 (38.46%)	
**Demographic factors**				
**Age**, years (median (IQR))	64.00 (56.00–70.00)	63.50 (57.50–68.00)	64.00 (56.00–71.00)	0.82
10–30	2 (3.08%)	0 (0.00%)	2 (8.00%)	0.17
30–50	6 (9.23%)	4 (10.00%)	2 (8.00%)	
50–70	39 (60.00%)	27 (67.50%)	12 (48.00%)	
70–85	18 (27.69%)	9 (22.50%)	9 (36.00%)	
**Gender**				1.00
Men (*n*, %)	41 (63.08%)	25 (62.50%)	16 (64.00%)	
Women (*n*, %)	24 (36.92%)	15 (37.50%)	9 (36.00%)	
**Comorbidities** (*n*, %)				
Diabetes mellitus	8 (12.31%)	5 (12.50%)	3 (12.00%)	1.00
Immunosuppression	11 (16.92%)	8 (20.00%)	3 (12.00%)	0.51
Tobacco use	35 (53.85%)	24 (60.00%)	11 (44.00%)	0.31
Charlson comorbidity score (median (IQR))	4.00 (3.00–5.00)	4.00 (3.00–5.00)	4.00 (2.00–6.00)	0.27
Previous H&N surgery	15 (23.08%)	12 (30.00%)	3 (12.00%)	0.13
Previous H&N RT	20 (30.77%)	17 (42.50%)	3 (12.00%)	**0.013**
Admission in the previous 2 years	19 (29.23%)	14 (35.00%)	5 (20.00%)	0.27
Resistant microorganisms	7 (10.77%)	4 (10.00%)	3 (12.00%)	1.00
**Blood parameters**^1^ (*n*, %)				
Anaemia ^2^	11 (16.92%)	10 (25.00%)	1 (4.00%)	**0.040**
Hypoalbuminemia ^3^	7 (10.77%)	4 (10.00%)	3 (12.00%)	1.00
WBC				0.31
<4 × 10^9^/L	3 (4.69%)	2 (5.13%)	1 (4.00%)	
4–11.3 × 10^9^/L	37 (56.92%)	20 (50.00%)	17 (68.00%)	
>11.3 × 10^9^/L	25 (39.06%)	18 (46.15%)	7 (28.00%)	
PCR				1.00
<0.5 mg/L	24 (36.92%)	15 (37.50%)	9 (36.00%)	
>0.5 mg/L	41 (65.08%)	25 (62.50%)	16 (64.00%)	
**Nutritional status**^1^ (*n*, %)				0.44
BMI <18.5	3 (4.62%)	2 (5.00%)	1 (4.00%)	
BMI 18.5–25	27 (41.54%)	19 (47.50%)	8 (32.00%)	
BMI >25	35 (53.85%)	19 (47.50%)	16 (64.00%)	
**Surgery**				
**Diagnosis** (*n*, %)				**0.028**
Benign tumours	4 (6.15%)	1 (2.50%)	3 (12.00%)	
Primary malignant tumours	41 (63.08%)	22 (55.00%)	19 (76.00%)	
Secondary malignant tumours	8 (12.31%)	7 (17.50%)	1 (4.00%)	
Salvage surgeries	7 (10.77%)	7 (17.50%)	0 (0.00%)	**0.038**
ORN Deformities	5 (7.69%)	3 (7.50%)	2 (8.00%)	
**ASA** (*n*, %)				0.15
1	5 (7.69%)	1 (2.50%)	4 (16.00%)	
2	27 (41.54%)	16 (40.00%)	11 (44.00%)	
3	31 (47.69%)	21 (52.50%)	10 (40.00%)	
4	2 (3.08%)	2 (5.00%)	0 (0.00%)	
**Surgery duration**, min (median [IQR])	600.00 (510.00–670.00)	657.50 (532.50–705.00)	555.00 (480.00–600.00)	**0.034**
**Tracheostomy** (*n*, %)	41 (63.08%)	29 (72.50%)	12 (48.00%)	**0.060**
**Neck dissection** (*n*, %)	42 (64.62%)	26 (65.00%)	16 (64.00%)	1.00
**Side**				0.73
Unilateral	29 (69.05%)	17 (65.38%)	12 (75.00%)	
Bilateral	13 (30.95%)	9 (34.62%)	4 (25.00%)	
**Type**				0.43
Selective levels I-III	30 (71.43%)	20 (76.92%)	10 (62.50%)	
Functional	8 (19.05%)	5 (19.23%)	3 (18.75%)	
Radical	1 (2.38%)	0 (0.00%)	1 (6.25%)	
Modified radical	3 (7.14%)	1 (3.85%)	2 (12.50%)	
**H&N resection** (*n*, %)				0.43
Oral cavity	50 (76.92%)	31 (77.50%)	19 (76.00%)	
Oropharynx	5 (7.69%)	4 (10.00%)	1 (4.00%)	
Skull base	4 (6.15%)	2 (5.00%)	2 (8.00%)	
Skin	4 (6.15%)	1 (2.50%)	3 (12.00%)	
Paranasal Sinuses	2 (3.08%)	2 (5.00%)	0 (0.00%)	
**Free flap** (*n*, %)				0.73
ALT	22 (33.85%)	13 (32.50%)	9 (36.00%)	
Forearm	19 (29.23%)	12 (30.00%)	7 (28.00%)	
Fibula	19 (29.23%)	13 (32.50%)	6 (24.00%)	
Other	5 (7.69%)	2 (5.00%)	3 (12.00%)	
**NNIS** (*n*, %)				0.52
1	30 (46.15%)	17 (42.50%)	13 (52.00%)	
2	33 (50.77%)	21 (52.50%)	12 (48.00%)	
3	2 (3.08%)	2 (5.00%)	0 (0.00%)	
**Reoperation <72 h** (*n*, %)	13 in 13 patients (20.00%)	11 in 11 patients (27.50%)	2 in 2 patients (8.00%)	0.128
Anastomoses	8 (12.30%)	8 (20.00%)	0 (0.00%)	**<0.001**
Flap loss	7 (87.50%)	7 (87.50%)	0 (0.00%)	
Flap survival	1 (12.50%)	1 (12.50%)	0 (0.00%)	
Hematoma drainage	5 (7.69%)	3 (7.50%)	2 (8.00%)	1.00
Flap loss	1 (20.00%)	1 (33.33%)	0 (0.00%)	
Flap survival	4 (80.00%)	2(66.67%)	2 (100.00%)	
**Flap loss** (*n*, %)	8 (12.30%)	8 (20.00%)	0 (0.00%)	**0.038**
**Flap survival** (*n*, %)	5 (7.69%)	3 (7.50%)	2 (8.00%)	1.00
**Complications**				
**Reoperation >72 h** (*n*, %)	23 in 17 patients (26.15%)	21 in 15 patients (37.50%)	2 in 2 patients (8.00%)	**0.009**
Dehiscence +/− fistula closure	10 (43.48%)	9 (42.86%)	1 (50.00%)	
Abscess drainage	7 (30.43%)	7 (33.33%)	0 (0.00%)	
Donor site	2 (8.70%)	2 (9.52%)	0 (0.00%)	
Others	4 (17.39%)	3 (14.29%)	1 (50.00%)	
**Duration of hospital admission**, days (median (IQR))	19.00 (12.00–26.00)	23.00 (18.00–34.50)	11.00 (10.00–16.00)	**<0.001**
(0–30 days)	53 (81.54%)	29 (72.50%)	24 (96.00%)	
1–7 days	2 (3.08%)	0 (0.00%)	2 (8.00%)	
8–14 days	20 (30.77%)	4 (10.00%)	16 (64.00%)	
15–21 days	17 (26.15%)	13 (32.50%)	4 (16.00%)	
22–30 days	14 (21.54%)	12 (30.00%)	2 (8.00%)	
(30–60 days)	7 (10.77%)	6 (15.00%)	1 (4.00%)	
(60–90 days)	4 (6.15%)	4 (10.00%)	0 (0.00%)	
>90 days	1 (1.54%)	1 (2.50%)	0 (0.00%)	
**Readmissions** (*n*, %)	9 (14.29%)	7 (17.95%)	2 (8.33%)	0.46
**Radiotherapy**	33 (50.77%)	22 (55.00%)	11 (44.00%)	0.45
Delay of RT (*n*, %)	20 (60.61%)	17 (77.27%)	3 (27.27%)	**0.009**
Time from surgery to initiation of RT, weeks (median (IQR))	8.00 (6.00–9.00)	8.00 (7.50–10.50)	6.00 (5.00–7.00)	**0.002**
**Mortality <6 months** (n, %)	2/28 (7.14%)	2/17 (7.69%)	0/11 (0.00%)	0.50
**Overall complications**^4^ (n, %)				**<0.0001**
≥1	47 (72.31%)	39 (97.50%)	8 (32.00%)	
0	18 (27.69%)	1 (2.50%)	17 (68.00%)	

HAIs, healthcare-associated infections; IQR, interquartile range; H&N, head and neck; RT, radiotherapy; WBC, white blood cells; PCR, C-reactive protein; BMI, body mass index; ORN, osteoradionecrosis; ASA, American Society of Anaesthesiologists; ALT, anterolateral thigh; NNIS, National Nosocomial Infections Surveillance; h, hours. ^1^ Presurgical parameters. ^2^ Anaemia: defined as haemoglobin <11.4 g/dL in females and <13 g/dL in males. ^3^ Hypoalbuminemia: defined as albumin <3.5 g/dL. ^4^ Postoperative complications previously analysed, not including HAIs and free-flap failure.

**Table 2 cancers-13-02109-t002:** Healthcare-associated infections in patients after head and neck free-flap reconstruction.

HAIs	69 (in 40 Patients)	%	% of Total Cohort (65 Patients)
**SSI**	36 (in 28 patients)	52.18	43.10
(A) Recipient SSI	32	46.38	
Head	5	7.24	
Neck	25	36.23	
Oral cavity	2	2.89	
(B) Donor SSI	4	5.80	6.15
Undergone surgery	1	1.45	
Conservative treatment	3	4.35	
**Respiratory infection**	16 (in 16 patients)	23.19	24.61
Nosocomial pneumonia	9	13.04	
Tracheobronchitis	7	10.14	
**Bloodstream infection**	9 (in 9 patients)	13.04	13.84
Sepsis	1	1.45	
Without sepsis	8	11.59	
**UTI**	4 (in 4 patients)	5.80	6.15
**Others**	4 (in 4 patients)	5.80	6.15

HAIs, health care-associated infections; SSI, surgical site infection; UTI, urinary tract infection.

**Table 3 cancers-13-02109-t003:** Microorganisms isolated from infected fields.

Microorganisms	N (71)	% of HAIs (69)
Usual microbial flora of the oral cavity	14	20.29
***Staphylococcus* spp.**	11	15.94
Methicillin-resistant *S. aureus* (MRSA)	7	10.14
Methicillin-sensitive *S. aureus* (MSSA)	1	1.45
*S. epidermidis*	3	4.35
***Pseudomonas aeruginosa***	12	17.39
*Pseudomonas aeruginosa*	5	7.25
MDR *Pseudomonas aeruginosa*	7	10.14
***Klebsiellas* spp.**	12	17.39
*Klebsiella pneumoniae*	4	5.8
MDR *Klebsiella pneumoniae*	5	7.25
*Klebsiella aerogenes*	2	2.9
MDR *Klebsiella aerogenes*	1	1.45
***Enterobacter* spp.**	9	13.04
*Enterobacter cloacae*	5	7.25
MDR *Enterobacter cloacae*	4	5.8
***Candida* spp.**	4	5.8
*Candida Albicans*	3	4.35
*Candida Krusei*	1	1.45
***Serratia marcescens* spp.**	2	2.89
*Serratia marcescens*	1	1.45
*MDR Serratia marcescens*	1	1.45
*Herpes simplex Virus (HSV)*	2	2.89
*Clostridium difficile toxin* detection in faeces	2	2.89
*Streptococcus pneumoniae*	1	1.45
*Escherichia Coli BLEE*	1	1.45
*SARS-CoV-2*	1	1.45

*S, Staphylococcus;* MDR, multidrug-resistant; BLEE, Extended-spectrum beta-lactamase-producing.

**Table 4 cancers-13-02109-t004:** Uni- and multivariate analyses of significant risk factors predicting HAIs.

Significant Risk Factors	Univariate		Multivariate	
	OR (95%CI)	*p*-Value	OR (95%CI)	*p*-Value
Previous H&N RT	5.42 (1.39–21.10)	0.013	5.57 (1.41–22.69)	0.014
Anaemia ^1^	8.00 (0.96–66.95)	0.040	-	-
Salvage surgery	-	0.038	-	-
Tracheostomy	2.86 (1.01–8.14)	0.065	-	-
Surgery duration, min (median (IQR))	1.01 (1.00–1.02)	0.034	1.01 (1.001–1.01)	0.048
Reoperation <72 h: Anastomoses	-	<0.001	-	-
Flap loss	-	0.038	-	-

HAIs, healthcare-associated infections; H&N, head and neck; RT, radiotherapy; h, hours; OR, Odds ratio; 95%CI, 95% confidence intervals. ^1^ Anaemia prior to surgery.

**Table 5 cancers-13-02109-t005:** Uni- and multivariate analyses of complications in patients with head and neck free-flap reconstruction.

Significant Complications	Univariate		Multivariate	
	OR (95%CI)	*p*-Value	OR (95%CI)	*p*-Value
Reoperation >72 h	6.89 (1.42–33.51)	0.009	-	-
Duration of hospital admission	1.16 (1.06–1.27)	<0.001	1.16 (1.05–1.28)	0.002
Radiotherapy				
Delay of RT	9.07 (1.72–47.67)	0.009	12.00 (1.89–76.38)	0.008
Time from surgery to RT	2.94 (1.25–6.93)	0.002	-	-
≥1 complication ^1^	17.34 (2.09–143.28)	<0.001	-	-

h, hours; RT, radiotherapy; OR, Odds ratio; 95%CI, 95% confidence intervals. ^1^ Postoperative complications previously analysed, not including HAIs and free-flap failure.

## Data Availability

The data presented in this study are available on request from the corresponding author, A.R.-Z. The data are not publicly available due to restrictions in accordance with consent provided by participants on the use of confidential data.

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
