# Peer review of "The Impact of Healthcare-Associated Infections in Patients Undergoing Oncological Microvascular Head and Neck Reconstruction: A Prospective Multicentre Study"

_cancers, 2021, doi:10.3390/cancers13092109_

Round 1
Reviewer 1 Report
Ana et al. conducted a prospective observational study of head and neck reconstruction in three tertary medical centers in Spain and wrote a paper on complications, including flap necrosis, antibiotic administration, and testing for organisms responsible for the infection.
Unfortunately, antibiotic administration and causative organisms were not statistically significant, and only known parameters such as previous radiation therapy were significant.
Similar to previous reports, the authors claim, infections occurred in 61.5% of cases, and complications of vascular anastomosis occurred in 8 of 65 cases. Unfortunately, the incidence of thrombosis at the anastomosis site is now considered to be about 5% by skilled plastic surgeons, so it must be said that many of the procedures in this cohort were inexperienced in terms of technique.
In view of the above, we believe that the paper does not contain useful information for many surgeons and is not worthy of acceptance.
Reviewer 2 Report
The authors did a legit job focusing on a growing issue all over the world. The aim is precise, and the introduction leads to the core of the study. However, few clarifications and correction are needed:
-Have patients been submitted to surgery after multidisciplinary team discussion and preoperative counseling?
-Line 108: "The aetiology was from benign or malignant tumours, or 108 osteoradionecrosis (ORN) deformities." This sentence is purely translated from Spanish, needs to be rephrased
-Line 268, "corresponding to receptor SSI", which does not have sense in English, is called the "recipient" site.
-Line 287 "The most commonly used antibiotics for the HAIs treatment were meropenem, piperacillin-tazobactam, amoxicillin-clavulanic acid and ciprofloxacin". Please report the posology you used, that could be useful for other authors
-Line 397, which is in my mind the core of the study: "Therefore, the duration of our antibiotic prophylaxis regimen is based on the diagnosis prior to surgery, since generally the surgical complexity is linked to the diagnosis, and this in turn to the risk of acquiring the infection intraoperatively." I'm afraid I have to disagree with this paragraph; it is well known that ORN cases might be much more challenging than recurrent or secondary cases. ORN patients have the same problem related to previous RT, such as contracture, fibrosis, and often vessel depletion. Moreover, if the indication for surgery and free flap reconstruction for ORN is correct, the ORN should be 3-4 grade, which comes with fistula, through and through bone exposure, etc. The results might be biased by the small number of ORN patients (5).
-Line 413 "Finally, the complex anatomy of the oro-cervical region and the use of distant microvascular free flaps for the reconstruction, often do not allow water-tight closure of the surgical wound, this potentially leading to prolonged contamination and fistula formation, especially in defects associating intra-oral and 416 extra-oral communication." I can't entirely agree with this sentence. One of the aims of free flap reconstruction is a water-tight separation between the recipient site and the neck. This should be checked by the end of flap insetting, drowning the cavity with betadine to confirm that the wound is sealed. If this was not done, it might partially explain the high rate of flap loss, which should not be higher than 5% in a tertiary care center
The design of the study is biased on the presumption that different diagnoses need different prophylaxis.
I do not fully agree with it, or at least, this should be proven. For example, treating the entire cohort with the same prophylaxis and analyzing which patients have fewer HAIs. Another solution might be including two cohorts of patients: one treated with the same prophylaxis and the other with the experimental one. Since it is an ongoing study, the addition of a comparative cohort that received different and homogeneous prophylaxis could improve the power of the study significantly.
Many sentences seem to be literally translated from the native language of the author. Therefore, an English check by a native speaker is mandatory. The words, from a grammar point of view, are correct, but the syntax in several sentences is misleading
Reviewer 3 Report
Thisi is an interesting paper examining the impact of Healthcare-Associated Infections (HAI) in patients undergoing oncological microvascular head and neck reconstruction. The paper is overall well written and only few grammar revisions are required. Some concerns
1) the author should better define the HAI, highlighting for example the type and the frequence of appearance in the clinical practice.
2) The authors should specify that the sample is very heterogeneous with regard of diagnosis (oral cavity tumors are not the same of larynx), the stage (recurrent have worse prognosis id compared with early stage/locally advanced) and the treatments employed (adjuvant/neoadjuvant radiothetapy strongly influences the appearance of local infections
3) the paragraph "discussion" is too large and it should make the reader tired, so i suggest to shorten it
Round 2
Reviewer 1 Report
None.
Reviewer 2 Report
The authors followed the suggestions and made proper modifications
Reviewer 3 Report
The suggestions have been accepted by the authors, so the paper can be published in this form